# Fruit Extract of *Sechium chinantlense* (Lira & F. Chiang) Induces Apoptosis in the Human Cervical Cancer HeLa Cell Line

**DOI:** 10.3390/nu15030667

**Published:** 2023-01-28

**Authors:** Ana Rocío Rivera-Martínez, Itzen Aguiñiga-Sánchez, Jorge Cadena-Iñiguez, Isabel Soto-Cruz, Alberto Monroy-García, Guadalupe Gómez-García, Edgar Ledesma-Martínez, Benny Weiss-Steider, Edelmiro Santiago-Osorio

**Affiliations:** 1Hematopoiesis and Leukemia Laboratory, Research Unit on Cell Differentiation and Cancer, Faculty of High Studies Zaragoza, National Autonomous University of Mexico, Mexico City CP 09230, Mexico; 2Department of Biomedical Sciences, School of Medicine, Faculty of High Studies Zaragoza, National Autonomous University of Mexico, Mexico City CP 56410, Mexico; 3Innovation in Natural Resource Management, Postgraduate College, Campus San Luis Potosí, Salinas de Hidalgo, San Luis Potosí CP 78622, Mexico; 4Immunology and Cancer Laboratory, Oncology Research Unit, Oncology Hospital, National Medical Center (IMSS), Mexico City CP 06720, Mexico

**Keywords:** apoptosis, intrinsic signaling pathway, cervical cancer

## Abstract

*Sechium edule* (Cucurbitaceae) is a commercial species of chayote and is just one of several species in the genus Sechium, whose extracts inhibit proliferation in tumor cell lines. The capacity of the wild species *Sechium chinantlense* (SCH) as an antitumor agent is unknown, as is the mechanism of action. In the present study, HeLa cervical cancer and HaCaT normal cell lines were treated with SCH and cell proliferation was inhibited in both cell lines in a dose-dependent manner similar to the effect of the antineoplastic agent cisplatin (Cis). Additionally, SCH arrested cell cycle progression but only in HeLa cells and induced apoptosis, as shown by phosphatidylserine translocation and caspase-3 activation, while Cis did so in both cell lines. Exploration of the mechanism of action of SCH in HeLa cells suggests that apoptosis was mediated by the intrinsic signaling pathway since there was no activation of caspase-8, but there was a release of cytochrome-c. These findings suggest that the SCH extract has the potential to selectively kill tumor cells by promoting apoptosis, without harming nontumor cells.

## 1. Introduction

Cancer is a growing health problem worldwide; according to the statistics of the International Agency for Research on Cancer [AIIC] of 2020, it is estimated that there were more than 19 million cases worldwide and 9.9 million cancer deaths. The recurrence rate of cervical cancer is still high in developing countries, and this disease is the seventh most common cancer that affects the total population and the fourth most common in women worldwide. In 2020, there were 604,127 new cases and 341,831 cases of deaths [1]. Of the global cervical cancer deaths each year, 90% occur in low- and middle-income countries [2].

Some hallmarks of cancer are uncontrolled growth and apoptosis evasion [3], which are the major hurdles in defeating cancer. Apoptosis is the cell’s natural mechanism of programmed cell death and is mainly regulated by intrinsic or extrinsic pathways. In both pathways, apoptosis is carried out by caspases (cysteine aspartyl-specific proteases), which can function as initiators or executors that cleave target proteins, eventually leading to the death of the cell [4]. The pathways of apoptosis are highly regulated; the extrinsic pathway is death receptor (DR)-mediated apoptosis with a high concentration of caspase-6, 7, and 8 but the intrinsic pathway is mediated by cytochrome-c release from mitochondria and activation of caspase-9, which stimulates the effector caspase-3 [5].

Chemotherapy is used in patients with cervical cancer to suppress DNA duplication or damage DNA in order to kill cancer cells that divide rapidly, but it is also used as an addition to definitive locoregional treatments (surgery or radiotherapy) to improve patient outcome and as a palliative therapy for patients with recurrent or de novo metastatic disease. These treatment modalities are applied independent of the histology of the disease, which may impact prognosis as well as response to treatment [6]. The optimal approach to the treatment of locally advanced cervical cancer is concurrent chemotherapy with radiotherapy (CCRT). The benefit of adding concurrent chemotherapy to radiation therapy (RT) is greater in earlier stages of the disease. Cisplatin is the most preferred agent for concurrent chemoradiotherapy, other agents have been tried for CCRT, but none have been found to be as effective or superior to cisplatin. In patients who cannot tolerate cisplatin, 5-FU (fluorouracil) is an alternative. The therapy for cervical cancer is changing with increasing knowledge regarding disease biology, particularly genomics and immunology. In locally advanced cervical cancers, CCRT with cisplatin remains the standard of care, but other drugs, such as bevacizumab, have been shown to improve survival. Immunotherapy agents such as pembrolizumab show promise in the treatment of advanced disease [7]. While concurrent chemoradiation has improved prognoses for women with locally advanced cervical cancer, outcomes remain poor for women with node-positive, recurrent, and metastatic disease [8]. Additionally, adverse side effects, such as bone marrow dysfunction inhibition, neurological, cardiac, pulmonary, and renal toxicity; and nausea, vomiting, and alopecia, reduce quality of life [9]. Efforts have always been focused on discovering new anticancer agents from medicinal plants and have successfully resulted in several experimental models using natural products for the treatment [10]. The fruit of *Sechium edule* (Jacq.) Sw, also known as “chayote”, is part of the human diet, but in ethnomedicine, it is also recommended for the treatment of cancer [9]. Several edible varietal groups have been described with different antiproliferative effects on different tumor cell lines [11], but the potential of the wild species *Sechium chinantlense* (Lira & F. Chiang) is unknown. Therefore, in this work, the antiproliferative and apoptosis-inducing potential of the *S. chinantlense* extract was compared in the human cervical HeLa cancer cell line and the nontumorigenic HaCaT epithelial cell line.

## 2. Materials and Methods 

### 2.1. Processing of Fruits and Preparation of Extract

The acquired *S. chinantlense* fruits were collected at horticultural maturity, 18 ± 2 days after the anthesis stage, by the Interdisciplinary Research Group on *Sechium edule* in Mexico, A. C. (GI*Se*M), at the Germplasm Bank, located in Veracruz, Mexico (19°49′ N; 97°21′ W), in 2013, and was authenticated by Cadena-Iñiguez J. at the National Seed Inspection and Certification Service (SNICS; Secretariat of Agriculture, Livestock, Rural Development, Fisheries and Food, Mexico).

The fruits were cut into flakes. In this procedure, the whole fruit was used, which also included parts of the exocarp and the seed. Later, the flakes were dried at 40 °C in an air circulation oven (BLUE-M, Electronic Company, Blue Island, IL, USA) until completely dehydrated. They were finally ground (in a common mill), to a particle size of 2 mm. Subsequently, a discontinuous extraction was carried out using 1.5 kg of the ground material, which was macerated in methanol at room temperature (20 ± 2 °C), and every 48 h, the solvent was changed. Finally the extract was concentrated in a rotary evaporator (IKA^®^ RV10, BUCHI R-114, Flawil, Switzerland) at 50 °C and the extract was recovered. The previous step was repeated 25 times until the solvent showed no color, and the extract was stored in an amber bottle, which was kept at room temperature. Of this extract, 71.2 mg was weighed in a tube and solubilized with 1 mL of PBS (phosphate buffer solution), and centrifuged at 2000 rpm for 5 min, and the supernatant was passed through a sterile filter and finally diluted in more PBS to obtain the following concentrations of the extract 0.0, 0.15, 0.3, 0.6, 1.2, 2.5, and 5 μg/mL. 

### 2.2. Cell Culture and Culture Conditions

In this study, 2 types of cells were used, HeLa, which is a cell line from an adenocarcinoma of human cervical epithelial tissue, and HaCaT, a keratinocyte cell line from nontumorigenic human skin epithelial tissue; both were obtained from the American Type Culture Collection (ATCC) (The Global Bioresource Center, Manassas, VA, USA). The HeLa and HaCaT cell lines were cultured at densities of 1 × 10^5^ and 7 × 10^4^ cells/mL, respectively, with reseeding every 48 h. They were maintained in glass Petri dishes (Pyrex, Corning, NY, USA) with Iscove’s Modified Dulbecco’s Medium (IMDM) culture medium (GIBCO-BRL Invitrogen, Grand Island, NY, USA) and supplemented with 10% fetal bovine serum (FBS) (Invitrogen GIBCO-BRL HyClone, Carlsbad, CA, USA) in an incubator (Thermo Forma, Marietta, OH, USA) at 37 °C with an atmosphere of 5% CO_2_ and 95% humidity [12]. All the experiments were carried out under these same culture conditions. Depending on the experiment, the appropriate concentrations of the *S. chinantlense* extracts or the control antineoplastic cisplatin (Cis) diluted in PBS (Blastolem RU^®^ 10 mg/10 mL, Teva Pharma AG, Basel, Switzerland) were used. For all experiments, 3 independent tests were performed with 3 repetitions per condition; in the case of histograms, a representative test is shown.

### 2.3. Antiproliferative Activity Assay

HeLa and HaCaT cells (2 × 10^4^ cells/well) were seeded onto 96-well plates (Corning Costar, St. Louis, MO, USA) and exposed for 72 h to 0.0, 0.15, 0.30, 0.6, 1.2, 2.5 or 5 μg/mL of extracts prepared from *S. chinantlense* or cisplatin as control antineoplastic agent. Next, they were fixed with 1.1% glutaraldehyde, stained with 0.1% crystal violet solution (Sigma-Aldrich, St. Louis, MO, USA), washed with distilled water, and solubilized with 10% acetic acid [13], after which the absorbance was measured at 570 nm using a plate reader (Tecan Spectra, Grödig, Austria). 

The results were plotted as percent proliferation relative to 0.0 μg/mL of extract. From these assays, the average inhibition concentration of cell proliferation by 50% (IC_50_) was determined, which was calculated using the linear regression equation and used in subsequent assays.

### 2.4. Discrimination of Apoptosis

The procedure was conducted according to the methodology provided in PE Annexin V Apoptosis Detection Kit (BD Pharmingen^TM^, Franklin Lakes, NJ, USA). Cells were incubated with the *S. chinantlense* extract at the IC_50_ value or with cisplatin for 72 h. All adhering and floating cells were harvested and then suspended in 1X binding buffer at a concentration of 1 × 10^5^ cells/mL. This cell suspension was transferred to an Annexin V-7AAD conjugate solution, and propidium iodide (PE) was added. The cells were incubated in the dark for 15 min at room temperature. The fluorescence of Annexin V-7AAD and PE (532 and 488 nm, respectively) was analyzed using a BD FACSAria II flow cytometer (BD Biosciences, Franklin Lakes, NJ, USA). 

### 2.5. Cell Cycle Analysis 

Cells were incubated with the *S. chinantlense* extract at the IC_50_ value or cisplatin for 72 h. All adhering and floating cells were harvested and washed twice with phosphate-buffered saline (PBS), then, 5 × 10^5^ cells were resuspended in ice-cold 70% ethanol and incubated at 4 °C for 12 h. Then, the cells were washed twice with PBS, and incubated with propidium iodide/RNAse solution was added to the cells and incubated in the dark for 1.0 h at room temperature before being analyzed by flow cytometry (BD FACSAria II Biosciences, NJ, USA). The analysis, histograms, and percentages of cell phases were obtained using Flowing Software version 2.5.1 (Perttu Terho, Turku Centre for Biotechnology University of Turku, Finland).

### 2.6. Caspase-8 Activity Assay

The procedure was conducted according to the methodology provided in the Caspase-8 Colorimetric Assay Kit (R&D Systems^®^, Minneapolis, MN, USA). Cells were incubated with the *S. chinantlense* extract at the IC_50_ value or cisplatin for 24 h. All adhering and floating cells were harvested, and 2 × 10^6^ cells were resuspended in an ice-cold lysis buffer for 10 min. Then, the cells were centrifuged, and the supernatant was transferred onto 96-well plates (Corning Costar, St. Louis, MO, USA) with 2X Reaction Buffer 8 and a colorimetric substrate of caspase-8. After 2 h at 37 °C, the absorbance was read at 490 nm using a microplate spectrophotometer (Thermo Scientific^TM^, Multiskan^TM^ GO, Madrid, Spain).

### 2.7. p53 and Caspase-3 Activity Assays 

HeLa and HaCaT cells were incubated with the *S. chinantlense* extract at IC_50_ value or cisplatin for 48 h. Then, all cells were harvested and washed with PBS. A total of 5 × 10^5^ cells were incubated with ice-cold BD Cytofix/Cytoperm^TM^ for 20 min and then washed twice with 1X BD Perm/Wash^TM^. The subsequent steps were performed according to the manufacturer’s instructions provided in the PE Active Caspase-3 Apoptosis Kit (BD Pharmingen^TM^ NJ, USA). Additionally, in a set-aside sample of cells, an Alexa Fluor^®^647 Mouse anti-p53 (ack382) antibody (BD Phosflow^TM^, Piscataway, NJ, USA) was used, which recognizes the acetylated lysine, residue 382, in the C-terminal region of p53. After 30 min in the dark, we washed both samples (the samples stained with caspase-3 PE and p53 Alexa Fluor^®^647) before analyzing them using a BD FACSAria II flow cytometer (Biosciences, NJ, USA). 

### 2.8. Cytochrome-c Release Assay

The concentration of cytochrome-c was determined by using the Human Cytochrome-c Immunoassay Quantikine^®^ ELISA (R&D Systems, Inc, Minneapolis, MN, USA). Cytochrome-c measurement was performed according to the manufacturer’s instructions. HeLa and HaCaT cells were incubated with the *S. chinantlense* extract at the IC_50_ value or cisplatin for 24 h. Then, 1 × 10^6^ cells were washed twice with PBS and incubated in lysis buffer for 1 h at room temperature with gentle shaking. Subsequently, the cells were centrifuged, and the supernatant was transferred onto 96-well plates (Corning Costar, St. Louis, MO, USA) with RD5P (1:10) diluent calibrator and after 2 h at room temperature, all samples were washed four times, and human cytochrome-c conjugate was added. After 2 h at room temperature and another four rounds of washing, we added substrate solution and incubated the mixture for 30 min in the dark at room temperature. The absorbance was read at 490 nm using a microplate spectrophotometer (Thermo Scientific^TM^, Multiskan^TM^ GO, Madrid, Spain).

### 2.9. Statistical Analysis

Values are expressed as the mean ± S.D. for control and experimental samples and statistical analysis was performed using one-way analysis of variance (ANOVA) followed by Tukey’s test. For this analysis, IBM SPSS Statistics Software Package version 18.0 (Armonk, NY, USA) was used. The values were considered as statistically significant if the *p* value was equal to or less than 0.05.

## 3. Results

### 3.1. Sechium chinantlense Extract Causes Dose-Dependent Inhibition of the Proliferation of HeLa and HaCaT Cell Lines

We evaluated the effect of *S. chinantlense* extract on a human cervical cancer cell line, HeLa, and a nontumorigenic epithelial cell line, HaCaT, using a crystal violet assay. The results showed a dose-dependent effect on cell proliferation of both cell lines upon treatment with increasing concentrations of *S. chinantlense*, and from 2.5 µg/mL of *S. chinantlense* extract, proliferation was inhibited by approximately 75% in both cell lines (Figure 1). Additionally, similar results were obtained when 0.3 µg/mL and 0.6 µg/mL of the antineoplastic agent cisplatin were used in HeLa and HaCaT cells, respectively. Using these dose-response results, the mean inhibition concentration (IC_50_) of *S. chinantlense* extract and cisplatin was calculated (Table 1), which was used in subsequent assays.

### 3.2. Effects of S. chinantlense Extract on the Cell Cycle of HeLa and HaCaT Cell Lines

Since *S. chinantlense* extract and cisplatin inhibited cell proliferation, the effect of each respective IC_50_ on cell cycle progression in HeLa and HaCaT cell lines was evaluated. The evaluation indicated that *S. chinantlense* extract induced an increase in the percentage of cells in the SubG1 phase in HeLa cells and the percentage of cells in S phase in HaCaT cells, while cisplatin increased the S and SubG1 phases in both cell lines (Figure 2).

### 3.3. S. chinantlense Extract Increased the Presence of Active p53 in HeLa Cells

The arrest of HeLa cells in SubG1 phase in the cell cycle suggests DNA damage and this event may be related to p53 activation. The incubation of both cell lines in the presence of the *S. chinantlense* extract or cisplatin at IC_50_ showed that both treatments increased the presence of the active form of p53 in both cell lines, but while the *S. chinantlense* extract had a greater effect on HeLa tumor cells (16.3% vs. 6.9% in HaCaT cells), cisplatin had a greater effect on nontumorigenic HaCaT cells (29.7% vs. to 9.2% for HeLa cells) (Figure 3). 

### 3.4. S. chinantlense Extract Induces Apoptosis in HeLa Cells

To determine if the *S. chinantlense* extract could induce apoptosis, we used annexin V-FITC and PI staining, and the stained cells were analyzed by flow cytometry. The translocation of phosphatidylserine, indicated by annexin V coupled to FITC, showed that exposure of HeLa cells to *S. chinantlense* extract increased early apoptotic cells (lower-right quadrants) to 51.7% vs. 5.8% in the control and increased late apoptotic cells (upper -right quadrants) to 41.2% vs. 10.7% in the control; in contrast, this effect was minimal in HaCaT cells, with early apoptotic cells to 2.86% vs. 1.64% in the control and late apoptotic cells to 5.80 vs. 5.26 in the control. Cisplatin induced apoptosis in both cell lines (Figure 4). These results suggested that the *S. chinantlense* extract could be effective in remarkably increasing the apoptosis of HeLa cancer cells at the early and late stages.

### 3.5. Active Caspase-8 Is Not Increased in HeLa or HaCaT Cells Treated with S. chinantlense Extract

Caspase-8 is considered a crucial modulator of the extrinsic pathway of apoptosis. As shown in Figure 5, *S. chinantlense* extract did not significantly increase caspase-8 activity in HeLa or HaCaT cells, after cisplatin treatment for 24 h, caspase-8 activity increased in HeLa cells but not in HaCaT cells.

### 3.6. S. chinantlense Extract Induces the Release of Cytochrome c in HeLa and HaCaT Cells

In the intrinsic pathway, p53 is related to cytochrome-c release from mitochondria. Here, both *S. chinantlense* extract and cisplatin increased the release of cytochrome c in both HeLa and HaCaT cell lines (Table 2), suggesting that apoptosis activation occurred through the intrinsic pathway, even in HaCaT cells, which exhibited a small percentage of cell death when treated with *S. chinantlense* extract.

### 3.7. S. chinantlense Extract Induces an Increase in Active Caspase-3 in HeLa Cancer Cells but Not in Nontumorigenic HaCaT Cells

Caspase-3 is responsible for most of the apoptotic effects, and upon activation, it can induce cleavage and DNA breakage and finally lead to the death of cells. Cisplatin induced caspase-3 activation in both HeLa and HaCaT cells (12.7% and 16.6%, respectively) while exposure to the *S. chinantlense* extract induced the activation of caspase-3 in the HeLa cancer cell line but not in the HaCat cell line (30.3% and 6.3%, respectively) (Figure 6). 

## 4. Discussion

This study shows that *S. chinantlense* extract inhibits proliferation in the cancer cell line HeLa and in the nontumorigenic cell line HaCaT. Additionally, the extract increased the percentage of cells in the SubG1 phase, translocation of phosphatidylserine (PS), release of cytochrome-c, and activation of p53 and caspase-3 (Figure 2, Figure 3, Figure 4 and Figure 6; Table 2. All these events are characteristic of cells entering apoptosis [14]. The data obtained in this study show that cisplatin induces the same biological response in HeLa cells, which has been widely reported [15,16]. These results allow us to suggest that the *S. chinantlense* extract induces death by apoptosis in HeLa tumor cells in the same manner as the antineoplastic agent cisplatin. Additionally, as has been previously reported, cisplatin is cytotoxic in HeLa cells but also in HaCaT cells [16]. In contrast, in HaCaT cells, the *S. chinantlense* extract induces the release of cytochrome c and arrest of the cell cycle in S phase but does not induce PS translocation or activate p53 or caspase-3, suggesting that *S. chinantlense*, although it has an antiproliferative effect, does not induce apoptosis in the nontumorigenic HaCaT cell line, but it does in HeLa cells. This differential effect has already been reported before. *Allium atroviolaceum* extract induces a cytotoxic effect on HeLa cells without affecting the normal cell line 3T3 [17], and resveratrol induces cell death in leukemic cells via caspases without affecting normal lymphocytes [18]. Most likely, what prevented HaCaT cells exposed to *S. chinantlense* extract from presenting the same apoptotic events, may be due to the presence of the inhibitor of apoptosis (IAP), since it has been shown that IAPs can inhibit apoptosis despite the release of cytochrome-c [19]. 

In some tumor cell lines, the release of cytochrome-c, a promoter of apoptosis via the intrinsic pathway, is promoted by plant phytochemicals [20]. Examples include lycopene, an abundant carotenoid phytochemical in tomatoes [21]; capsaicin, a phenolic compound from chili peppers (*Capsicum* sp.), that induces a rapid increase in reactive oxygen species followed by a disruption of mitochondrial membrane potential [22]; luteolin and curcumin found in celery and turmeric, respectively, which depolarize mitochondria and induce apoptosis in human tumor cells [23,24]; the crude extract of *A. atroviolaceum*, which induces the activation of caspase-9 and caspase-3. These examples all indicate that the mitochondrial pathway is involved in apoptosis induction in HeLa cells by plant phytochemicals [17]. A preliminary analysis of the phytochemical content in the *S. chinantlense* extract revealed that the main components are terpenes, flavonoids (flavones and flavonols), saponins and tannins [25] and that other phytochemicals reported in the species *S. edule*, such as peroxidases, sterols, phenols, polyphenols, fatty acid esters, carotenoids, vitamin c, antioxidants, and cucurbitacins, are also present [5,26,27,28,29,30]. This set of phytochemicals has been isolated individually from other plants and is reported to have anti-allergenic, anti-inflammatory, antimutagenic, antioxidant, antiviral, and antitumor and antileukemic effects [5,27,31,32,33,34,35,36,37], which allows us to assume that *S. chinantlense*, as it contains some of these compounds, such as cucurbitacins, is responsible for its antineoplastic activity.

It has also been reported that cucurbitacins show anticancer activity [29,38,39,40,41,42,43,44,45,46], via induction of cytochrome-c release, activation of caspase-3 and caspase-9, and apoptosis through the intrinsic pathway [47], and there are no reports that show that they activate the death receptor of the extrinsic pathway. Cucurbitacins have been present in the Cucurbitaceae family and widely in some of its genera, such as Sechium, to which *S. chinantlense* belongs [47]. Therefore, the apoptotic effect of the *S. chinantlense* extract may occur through the intrinsic pathway activated by cucurbitacins, as the main phytochemical responsible for the antineoplastic activity of the *S. chinantlense* extract. If so, it could serve as an adjuvant and complementary treatment, to increase the therapeutic efficacy of antineoplastic drugs and reduce the toxicity generated to the organs [48] and, therefore, reduce the side effects derived from chemotherapy treatment. Therefore, it is necessary to continue this research using in vivo models.

As we mentioned before, many plant species have been shown to have antiproliferative effects by using high concentrations of extracts; however, *A. atroviolaceum* is among some of the notable exceptions with effectiveness at low concentrations but still requires an IC_50_ of 75 μg/mL to induce apoptosis in HeLa cells [17]. In this sense, *S. chinantlense* extract has a much higher antiproliferative potential since it has an IC_50_ 11 times lower than that required for an extract of oncological interest, according to the National Cancer Institute of the United States States [49]. It should be noted that the inhibition potential of *S. chinantlense* is similar to that of cisplatin, with a difference of 1.1 μg/mL between their IC_50_ values, and *S. chinantlense* has the advantage of being a natural extract that is considered less toxic to normal cells [24]. It should be noted that the *S. edule* wild varietal group extract has an IC_50_ of 1170 μg/mL [11], so the *S. chinantlense* extract is 600 times more effective as an antitumor agent, with similar in effectiveness in inhibiting the proliferation of the same cervical cancer line to that reported in a new cultivar of *S. edule* called “Perla Negra”, where the IC_50_ is 1.85 μg/mL [50]. Thus, the antitumor potential of the Sechium genus is evidenced by another study carried out with a hybrid of *S. edule* in leukemia cell lines, and the extract showed an antiproliferative and proapoptotic effect with an IC_50_ lower than 1.3 μg/mL [51]. Finally, in the WEHI-3 leukemic line, the *S. chinantlense* extract inhibited cell proliferation with an IC50 of 0.3 μg/mL, while 3.56 μg/mL was required to inhibit normal bone marrow cells [25], which correlates with our finding of the differential selective toxic effect against the HeLa tumor cell line compared to the nontumorigenic HaCaT cell line, so it would be important to analyze this differential effect between tumors cells and nontumorigenic cells in other types of tumor in the future. In the same way, in this study, the observation that *S. chinantlense* has a lower effect on normal cells indicates that it could also have fewer side effects, and it is necessary to perform more expansive experiments and even clinical studies to further explore the potential of *S. chinantlense* alone and in combination with the antineoplastic cisplatin agent.

## 5. Conclusions 

In this study, *Sechium chinantlense* extract shows antitumor activity similar to that of antineoplastic agent cisplatin in eliminating neoplastic cells by inducing apoptosis, but *S. chinantlense* extract has certain selectivity for cervical cancer cells over nontumorigenic keratinocytes, which indicates the value of continuing the biomedical investigation of this wild species of the Sechium genus.

## Figures and Tables

**Figure 1 nutrients-15-00667-f001:**
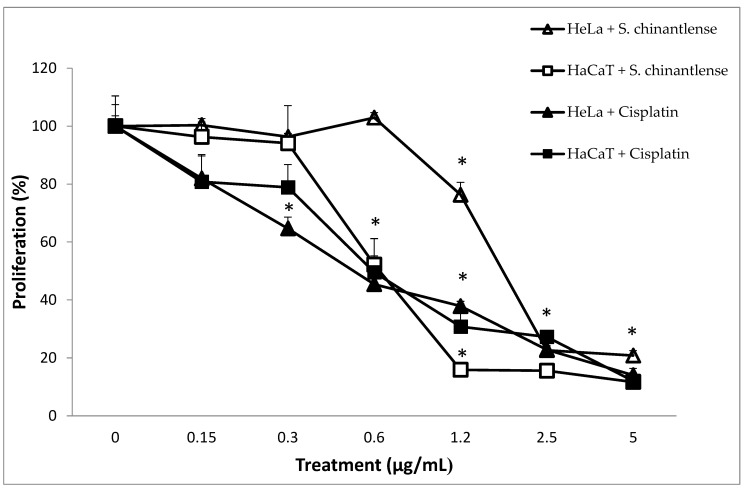
Effect of different *S. chinantlense* extract concentrations on the proliferation of HeLa and HaCaT cell lines. The proliferation of HeLa and HaCaT cells was evaluated after 72 h in the presence of *S. chinantlense* extract or cisplatin. Average values of three independent experiments ± S.D. * *p* < 0.01 ANOVA—Tukey’s test with respect to the control of each cell line.

**Figure 2 nutrients-15-00667-f002:**
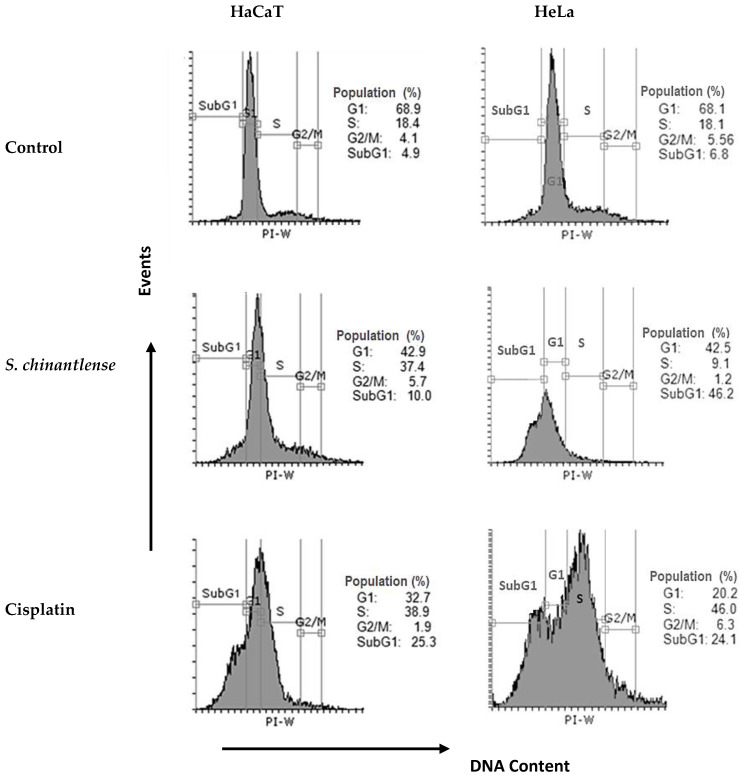
Effects of *S. chinantlense* extract or cisplatin at the IC_50_ value on cell cycle distribution in HeLa and HaCaT cells. Histograms represent flow cytometric analysis of untreated control, *S. chinantlense* extract-treated, and cisplatin-treated HeLa and HaCaT cells. The *X*-axis represents the intensity of PI staining, which is directly proportional to the amount of DNA in cells, and the *Y*-axis represents cell number.

**Figure 3 nutrients-15-00667-f003:**
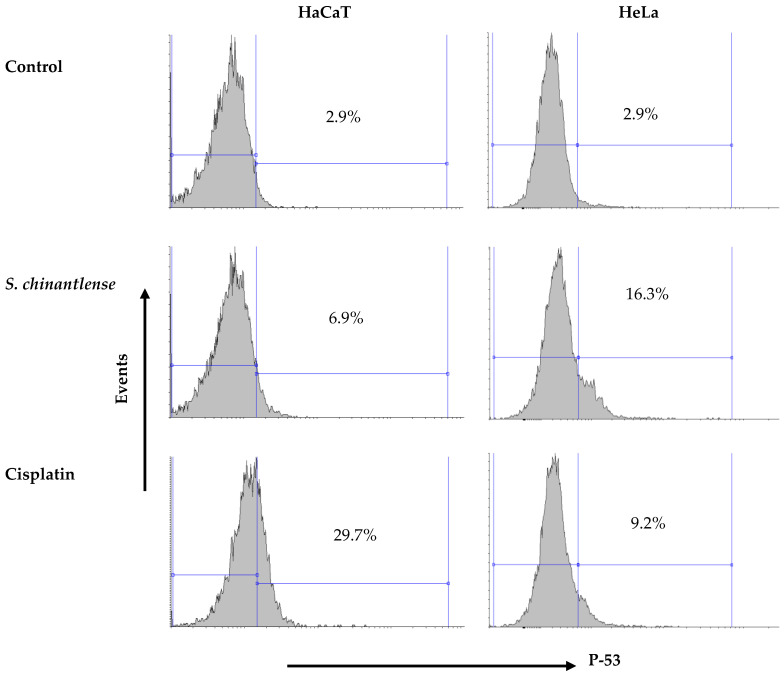
Effect of incubation with *S. chinantlense* extract or cisplatin at the IC_50_ value for 48 h on the level of active p53 in HeLa and HaCaT cell lines.

**Figure 4 nutrients-15-00667-f004:**
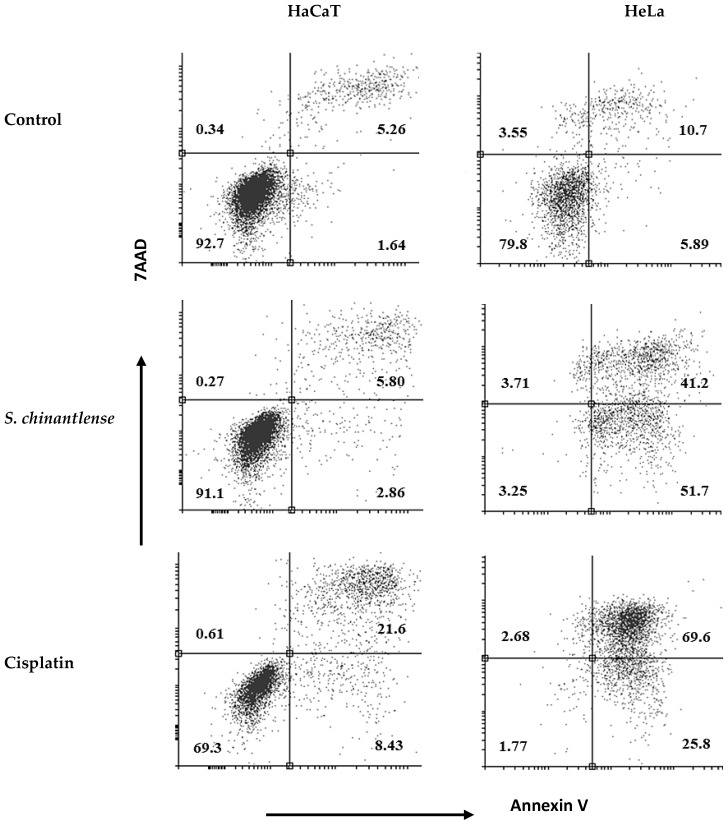
Apoptosis analysis upon treatment of HeLa and HaCaT cells with *S. chinantlense* extract at the IC_50_ value. The distribution of cells undergoing early and late apoptosis together with viable cells was determined at 72 h in comparison to control or cisplatin at the IC_50_ value, using Annexin V-FITC and propidium iodide flow cytometric analysis.

**Figure 5 nutrients-15-00667-f005:**
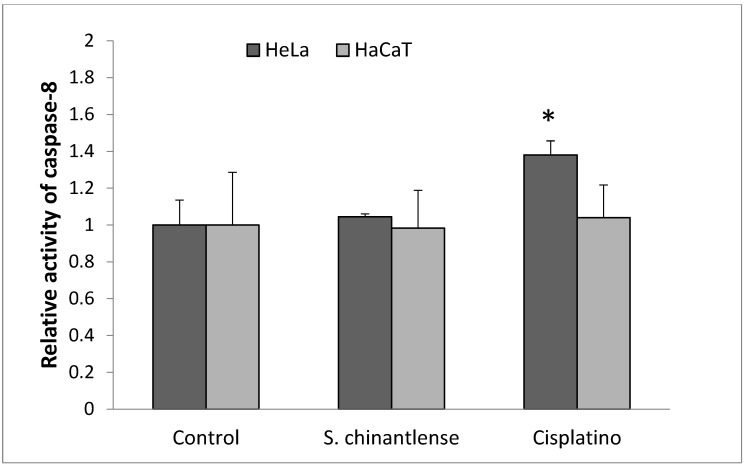
Activation of caspase-8 in HeLa cancer cells after incubation with *S. chinantlense* extract or cisplatin at the IC_50_ value for 24 h. The values are presented as the mean ± S.D., where ∗ indicates a significant difference relative to the control (*p* < 0.05).

**Figure 6 nutrients-15-00667-f006:**
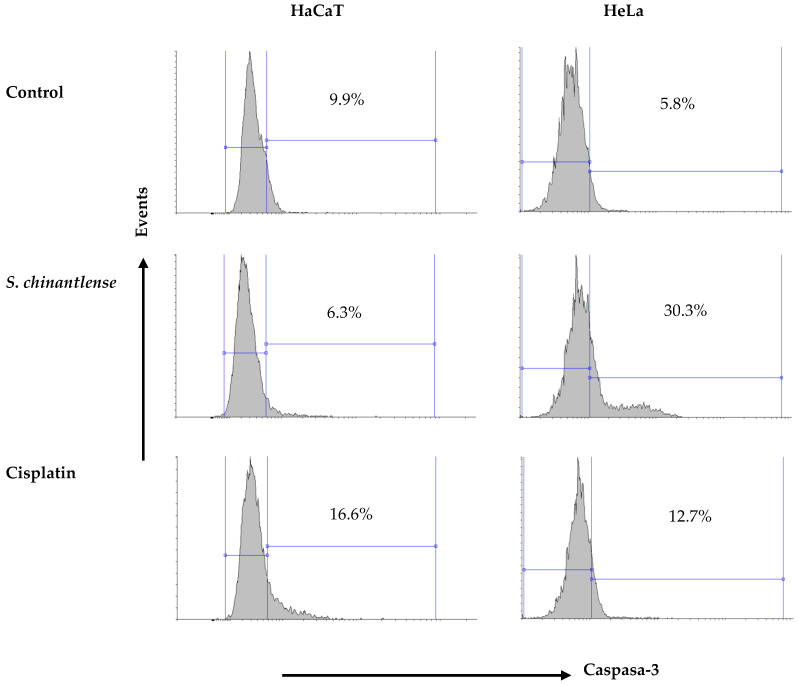
Activation of caspase-3 in HeLa and HaCaT cell lines after incubation with *S. chinantlense* extract or cisplatin at the IC_50_ value for 48 h.

**Table 1 nutrients-15-00667-t001:** Mean concentration of inhibition of cell proliferation [IC_50_] (µg/mL) induced by *S. chinantlense* extract or cisplatin in HeLa and HaCaT cell lines.

Cell Line	*S. chinantlense*	Cisplatin
	(μg/mL)	
HeLa	1.82	0.72
HaCaT	0.73	0.57

**Table 2 nutrients-15-00667-t002:** Mean concentration of cytoplasmic cytochrome c (μg/mL) in HeLa and HaCaT cell lines induced by *S. chinantlense* extract or cisplatin at the IC_50_ value for 24 h.

	Control	*S. chinantlense*	Cisplatin
		(μg/mL)	
HaCaT	27.7 ± 11.2	50.5 ± 2.3 *	47.7 ± 7.2 *
HeLa	28.0 ± 1.8	56.7 ± 17.2 *	54.7 ± 6.2 *

The values are presented as the mean ± S.D., where ∗ indicates a significant difference relative to the control (*p* < 0.05).

## Data Availability

The data presented in this study are available on request from the corresponding author.

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
