# Peer review of "Fruit Extract of Sechium chinantlense (Lira & F. Chiang) Induces Apoptosis in the Human Cervical Cancer HeLa Cell Line"

_nutrients, 2023, doi:10.3390/nu15030667_

Round 1
Reviewer 1 Report
The authors show that a fruit extract of Sechium chinantlense (Lira & F. Chian) induces apoptosis in the human cervical cancer HeLa cell line. In effect, it could be comparable to the antineoplastic agent cisplatin (Cis). However, the results show that the effect was dose-dependent.
Minor details:
1. Please, provide more information about: Cell culture and culture condition. Another media was used? under what conditions?
2. How many repetitions by condition were assessed? Why did you use parametric test over non-parametric test?
3. English revision by an expert is necessary to avoid grammatical errors. Please, review the grammatical errors.
4. It is necessary to review the wording in the following lines: 23-24; 36; 58-62; 66-68; 74; 95-96; 107-108; 113; 127-131; 138-141; 154-157; 252; 374-379; 387-398; 411-412; 414-419; 431-435.
Kind regards,
Author Response
- Please, provide more information about: Cell culture and culture condition. Another media was used? under what conditions?
We have expanded the information in the methodology section, we hope to be explicit enough.
(page 1 line 80-99)
Chemotherapy is used in patients with cervical cancer to suppress DNA duplication or damage DNA in order to kill cancer cells that divided rapidly, but it is also used as an addition to definitive locoregional treatments (surgery or radiotherapy) to improve patient outcome and as a palliative therapy for patients with recurrent or de novo metastatic disease. These treatment modalities are applied independent of the histology of the disease, which may impact prognosis as well as response to treatment [6]. The optimal approach to the treatment of locally advanced cervical cancer is concurrent chemotherapy with radiotherapy (CCRT). The benefit of adding concurrent chemotherapy to radiation therapy (RT) is greater in earlier stages of the disease. Cisplatin is the most preferred agent for concurrent chemoradiotherapy, other agents have been tried for CCRT, but none have been found to be as effective or superior to cisplatin. In patients who cannot tolerate cisplatin, 5-FU (fluorouracil) is an alternative. The therapy for cervical cancer is changing with increasing knowledge regarding disease biology, particularly genomics and immunology. In locally advanced cervical cancers, CCRT with cisplatin remains the standard of care, but other drugs, such as bevacizumab, have been shown to improve survival. Immunotherapy agents such as pembrolizumab show promise in the treatment of advanced disease [7]. While concurrent chemoradiation has improved prognoses for women with locally advanced cervical cancer, outcomes remain poor for women with node-positive, recurrent, and metastatic disease [8]. Additionally, adverse side effects, such as bone marrow dysfunction inhibition, neurological, cardiac, pulmonary, and renal toxicity; and nausea, vomiting, and alopecia, reduce quality of life [9]. Efforts have always been focused on discovering new anticancer agents from medicinal plants and have successfully resulted in several experimental models using natural products for the treatment [10].
(p2, line 101-115) .
In this study, 2 types of cells were used, HeLa, which is a cell line from an adenocar-cinoma of human cervical epithelial tissue, and HaCaT, a keratinocyte cell line from non-tumorigenic human skin epithelial tissue; both were obtained from the American Type Culture Collection (ATCC) (The Global Bioresource Center, Manassas, VA). The HeLa and HaCaT cell lines were cultured at densities of 1x105 and 7x104 cells/mL, respectively, with reseeding every 48 hours. They were maintained in glass Petri dishes (Pyrex, USA) with Iscove's Modified Dulbecco's Medium (IMDM) culture medium (GIBCO-BRL Invitrogen Grand Island, NY) and supplemented with 10% fetal bovine serum (FBS) (Invitrogen GIBCO-BRL HyClone, Carlsbad, CA) in an incubator (Thermo Forma, OH, USA) at 37 °C with an atmosphere of 5% CO2 and 95% humidity [12]. All the experiments were carried out under these same culture conditions. Depending on the experiment, the appropriate concentrations of the S. chinantlense extracts or the control antineoplastic cisplatin (Cis) diluted in PBS (Blastolem RU® 10 mg/10 mL, Teva Pharma AG, Switzerland) were used. For all experiments, 3 independent tests were performed with 3 repetitions per condition; in the case of histograms, a representative test is shown.
- How many repetitions by condition were assessed? Why did you use parametric test over non-parametric test?
In the statistical analysis (page 4 line 184) we mention that we use parametric tests (ANOVA). The data in which the ANOVA tukey tests were applied 1) meet the conditions required by the parametric tests (population size, normal distribution, homogeneity of variances, etc.) to determine differences between means 2) although data that meet the criterion of normality are amenable to analysis with non-parametric tests, it is preferred to use a parametric test over a non-parametric one because the former has more statistical power than the latter.
- English revision by an expert is necessary to avoid grammatical errors. Please, review the grammatical errors.
Thank you for your comment, we have submitted the manuscript for grammar review by an expert, we annex the evidence of the grammar revision.
- It is necessary to review the wording in the following lines: 23-24; 36; 58-62; 66-68; 74; 95-96; 107-108; 113; 127-131; 138-141; 154-157; 252; 374-379; 387-398; 411-412; 414-419; 431-435.
Thank you for your comment, we have submitted the manuscript for grammar review by an expert, we annex the evidence of the grammar revision.
Reviewer 2 Report
The main goal of the manuscript is to figure out how an alcohol extract made from Sechium chinantlense affects cervical tumor cells. The results are exciting and similar to what cisplatin did to cancer cells in terms of toxicity.
This topic is of interest to readers of the current journal. I would recommend that they explain more about the colorectal cancer cell and any clinical trials on any extracts in cervical cancer, as well as address cancer efficacy.In this way, it will improve the impact of proposed extracts to complement existing treatments in clinical use.
The technique proposed to prepare the extract, can be characterized for constituents like polypehols, terpenoids, proteins, and others.
Please keep in mind that the Cas9, 3, and p53 assays are critical in these cancer models, which is why the authors focused on molecular signaling. How do these signaling pathways help readers opt for this extract? Further research needs to be proposed. I would suggest putting it to use in the clinic and realizing that it needs to be used with cisplatin.
Author Response
I would recommend that they explain more about the colorectal cancer cell and any clinical trials on any extracts in cervical cancer, as well as address cancer efficacy.
About the colorectal cancer cell. Please take into consideration that in our manuscript we do not address colorectal cancer.
About any clinical trials on any extracts in cervical cancer, as well as address cancer efficacy. We have supplemented the information on cervical cancer treatment.
page 2 line 48-70.
Chemotherapy is used in patients with cervical cancer to suppress DNA duplication or damage DNA in order to kill cancer cells that divided rapidly, but it is also used as an addition to definitive locoregional treatments (surgery or radiotherapy) to improve patient outcome and as a palliative therapy for patients with recurrent or de novo metastatic disease. These treatment modalities are applied independent of the histology of the disease, which may impact prognosis as well as response to treatment [6]. The optimal approach to the treatment of locally advanced cervical cancer is concurrent chemotherapy with radiotherapy (CCRT). The benefit of adding concurrent chemotherapy to radiation therapy (RT) is greater in earlier stages of the disease. Cisplatin is the most preferred agent for concurrent chemoradiotherapy, other agents have been tried for CCRT, but none have been found to be as effective or superior to cisplatin. In patients who cannot tolerate cisplatin, 5-FU (fluorouracil) is an alternative. The therapy for cervical cancer is changing with increasing knowledge regarding disease biology, particularly genomics and immunology. In locally advanced cervical cancers, CCRT with cisplatin remains the standard of care, but other drugs, such as bevacizumab, have been shown to improve survival. Immunotherapy agents such as pembrolizumab show promise in the treatment of advanced disease [7]. While concurrent chemoradiation has improved prognoses for women with locally advanced cervical cancer, outcomes remain poor for women with node-positive, recurrent, and metastatic disease [8]. Additionally, adverse side effects, such as bone marrow dysfunction inhibition, neurological, cardiac, pulmonary, and renal toxicity; and nausea, vomiting, and alopecia, reduce quality of life [9]. Efforts have always been focused on discovering new anticancer agents from medicinal plants and have successfully resulted in several experimental models using natural products for the treatment [10].
In this way, it will improve the impact of proposed extracts to complement existing treatments in clinical use.
The story we present here with cell lines is the first phase of a much larger and more ambitious study. There are still many research questions that need to be answered before starting the clinical trial phase, but we believe that we are on the right path, and in the future, we will be able to present preclinical and clinical trials with SCH as we add in page 11, line 482-486.
In the same way, in this study, the observation that S. chinantlense has a lower effect on normal cells indicates that it could also have fewer side effects, and it is necessary to perform more expansive experiments and even clinical studies to further explore the potential of S. chinantlense alone and in combination with the antineoplastic cisplatin agent.
The technique proposed to prepare the extract, can be characterized for constituents like polypehols, terpenoids, proteins, and others.
It has been reported that the highest percentage of metabolites present in SCH extract is in a greater presence of terpenes, flavonoids, saponins, and tannins, this data has been reported and which is added in the discussion, on page 11 lines 439-441.
A preliminary analysis of the phytochemical content in the SCH revealed that the main components are terpenes, flavonoids (flavones and flavonols), saponins and tannins [25]
Please keep in mind that the Cas9, 3, and p53 assays are critical in these cancer models, which is why the authors focused on molecular signaling. How do these signaling pathways help readers opt for this extract? Further research needs to be proposed. I would suggest putting it to use in the clinic and realizing that it needs to be used with cisplatin.
We have supplemented the information in the discussion on page 12 line 482-486 as follows:
In the same way in this study, when observing that SCH has less effect on normal cells, it could be suggested that it would also reduce side effects, for which it is necessary to continue expanding the experiments, even reaching clinical studies with the option of using them together, the extract and the antineoplastic.
Reviewer 3 Report
The manuscript entitled by Rivera-Martínez et al., entitled: Fruit extract of Sechium chinantlense induces apoptosis in the human cervical cancer HeLa cell line documents the antiproliferative effect of SCH in HeLa cervical cancer and HaCaT normal keratinocytes cell lines. The manuscript is interesting but has a few concerns. Please find my comments below.
Is the antitumor effect of SCH cancer cell type specific? It would be ideal if authors could include the effect of SCH on other cancer cell types.
Authors should include the expression analysis at protein and/or RNA levels for the Apoptosis markers.
Line 47 needs a reference, and the authors should elaborate on the drawbacks of current treatments and How the SCH extract is more effective than the already existing clinically practiced chemotherapeutic agents?
Adding a few lines in the discussion about other known biological activity or SCH extracts will also be helpful.
Introduction, authors should add 2021 data if available.
Author Response
Is the antitumor effect of SCH cancer cell type specific? It would be ideal if authors could include the effect of SCH on other cancer cell types.
We have supplemented the information discussed on page 11 line 477-486 as follows:
Finally, in the WEHI-3 leukemic line, the S. chinantlense extract inhibited cell proliferation with an IC50 of 0.3 μg/mL, while 3.56 μg/mL was required to inhibit normal bone marrow cells [25], which correlates with our finding of the differential selective toxic effect against the HeLa tumor cell line compared to the nontumorigenic HaCaT cell line, so it would be important to analyze this differential effect between tumors cells and nontumorigenic cells in other types of tumor in the future. In the same way, in this study, the observation that S. chinantlense has a lower effect on normal cells indicates that it could also have fewer side effects, and it is necessary to perform more expansive experiments and even clinical studies to further explore the potential of S. chinantlense alone and in combination with the antineoplastic cisplatin agent.
Authors should include the expression analysis at protein and/or RNA levels for the Apoptosis markers.
The story we present here with cell lines is the first phase of a much larger and more ambitious study. We consider that at this time of the study, it is not pertinent to present said data and we prefer that the apoptosis induction as the biological effect of the extract be the central part of the article.
Line 47 needs a reference, and the authors should elaborate on the drawbacks of current treatments and How the SCH extract is more effective than the already existing clinically practiced chemotherapeutic agents?
This is an exploratory study, and the results obtained motivate us to continue with it, for which we hope in the future to have experimental elements that allow us to discern whether the SCH extract is more effective than existing chemotherapeutic agents. Meanwhile, we have updated the information regarding the clinical practice of cervical cancer on page 2 line 48-68.
Chemotherapy is used in patients with cervical cancer to suppress DNA duplication or damage DNA in order to kill cancer cells that divided rapidly, but it is also used as an addition to definitive locoregional treatments (surgery or radiotherapy) to improve patient outcome and as a palliative therapy for patients with recurrent or de novo metastatic disease. These treatment modalities are applied independent of the histology of the disease, which may impact prognosis as well as response to treatment [6]. The optimal approach to the treatment of locally advanced cervical cancer is concurrent chemotherapy with radiotherapy (CCRT). The benefit of adding concurrent chemotherapy to radiation therapy (RT) is greater in earlier stages of the disease. Cisplatin is the most preferred agent for concurrent chemoradiotherapy, other agents have been tried for CCRT, but none have been found to be as effective or superior to cisplatin. In patients who cannot tolerate cisplatin, 5-FU (fluorouracil) is an alternative. The therapy for cervical cancer is changing with increasing knowledge regarding disease biology, particularly genomics and immunology. In locally advanced cervical cancers, CCRT with cisplatin remains the standard of care, but other drugs, such as bevacizumab, have been shown to improve survival. Immunotherapy agents such as pembrolizumab show promise in the treatment of advanced disease [7]. While concurrent chemoradiation has improved prognoses for women with locally advanced cervical cancer, outcomes remain poor for women with node-positive, recurrent, and metastatic disease [8]. Additionally, adverse side effects, such as bone marrow dysfunction inhibition, neurological, cardiac, pulmonary, and renal toxicity; and nausea, vomiting, and alopecia, reduce quality of life [9].
Adding a few lines in the discussion about other known biological activity or SCH extracts will also be helpful.
To our knowledge, this is the first study regarding the biological effect of SCH in a cervical cancer model and we only have information on the antiproliferative effect of SCH in WEHI-3 leukemic cells, as we mentioned on page 11 line 477-4479. Although we could suggest that SCH could have a similar effect to other Sechium spp. extracts, we have no experimental evidence to support this assumption.
(11 line 477-4479)
Finally, in the WEHI-3 leukemic line, the S. chinantlense extract inhibited cell proliferation with an IC50 of 0.3 μg/mL, while 3.56 μg/mL was required to inhibit normal bone marrow cells [25],
Introduction, authors should add 2021 data if available.
We appreciate the suggestion, but unfortunately, the statistics of the International Agency for Research on Cancer on the WHO site until today, there is only the information for the year 2020, available in https://gco.iarc.fr/today/fact-sheets-cancers .
Reviewer 4 Report
Rivera-Martínez et al., reported “Fruit extract of Sechium chinantlense (Lira & F. Chian) induces apoptosis in the human cervical cancer HeLa cell line” is complete and well written. This study investigated the anticancer activity of methanol extract of Sechium chinantlense fruit. This study is well organised and presented.This study reported the crude extract of SCH, however they did not evaluate the extract's active component. Consequently, this MS has significant issues that must be addressed. Comments 1. What is the novelty of this study? 2. Why are these two cancer cell lines chosen? 3. What are the main active components in the methanolic fruit extract of Sechium chinensis? 4. To determine the main bioactive components, authors must conduct HPLC investigations. 5. Authors must provide microscopic images of malignancy and normal cell lines following treatments.
Author Response
Comments 1. What is the novelty of this study?
SCH is a wild species of the genus Sechium, which, although it is not edible, is characterized by having an antiproliferative effect superior to edible varieties. Also the observation that S. chinantlense has a lower effect on normal cells indicates that it could also have fewer side effects, but it is necessary to perform more expansive experiments and even clinical studies to further explore the potential of S. chinantlense as an antineoplastic agent.
In the discussion we extensively mentioned the advantage of SCH as a potential antitumor agent.
(page 11-12, line 461-486)
As we mentioned before, many plant species have been shown to have antiprolifera-tive effects by using high concentrations of extracts; however, A. atroviolaceum is among some of the notable exceptions with effectiveness at low concentrations but still requires an IC50 of 75 μg/mL to induce apoptosis in HeLa cells [17]. In this sense, S. chinantlense ex-tract has a much higher antiproliferative potential since it has an IC50 11 times lower than that required for an extract of oncological interest, according to the National Cancer Insti-tute of the United States States [49]. It should be noted that the inhibition potential of S. chinantlense is similar to that of cisplatin, with a difference of 1.1 μg/mL between their IC50 values, and S. chinantlense has the advantage of being a natural extract that is considered less toxic to normal cells [24]. It should be noted that the S. edule wild varietal group extract has an IC50 of 1170μg/mL [11], so the S. chinantlense extract is 600 times more effective as an antitumor agent, with similar in effectiveness in inhibiting the proliferation of the same cervical cancer line to that reported in a new cultivar of S. edule called “Perla Negra”, where the IC50 is 1.85 μg/mL [50]. Thus, the antitumor potential of the Sechium genus is evidenced by another study carried out with a hybrid of S. edule in leukemia cell lines, and the extract showed an antiproliferative and proapoptotic effect with an IC50 lower than 1.3 μg/mL [51]. Finally, in the WEHI-3 leukemic line, the S. chinantlense extract inhibited cell proliferation with an IC50 of 0.3 μg/mL, while 3.56 μg/mL was required to inhibit normal bone marrow cells [25], which correlates with our finding of the differential selective toxic effect against the HeLa tumor cell line compared to the nontumorigenic HaCaT cell line, so it would be important to analyze this differential effect between tumors cells and non-tumorigenic cells in other types of tumor in the future. In the same way, in this study, the observation that S. chinantlense has a lower effect on normal cells indicates that it could also have fewer side effects, and it is necessary to perform more expansive experiments and even clinical studies to further explore the potential of S. chinantlense alone and in combination with the antineoplastic cisplatin agent.
- Why are these two cancer cell lines chosen?
As we mentioned in the introduction (page 1 line 30) cancer is a global health problem, and cervical cancer is one of the types of cancer that most affects the total population and most frequently in women worldwide. in this sense, HeLa cell line is the main model in the study of this type of cancer, and HaCaT, the non-tumoral cell line most frequently used as a contrast for Hela cells.
- What are the main active components in the methanolic fruit extract of Sechium chinensis?
It has been reported that the highest percentage of metabolites present in the S. chinantlense extract is in a greater presence of terpenes, flavonoids, saponins and tannins, this data has been reported previously and is cited in the discussion, on page 11 line 439-441.
- To determine the main bioactive components, authors must conduct HPLC investigations.
The advance that we present here with cell lines is the first phase of a much larger and more ambitious study, so please consider our advance at this level. Once the relevant antineoplastic effect is shown, the next phase is what this rightly suggests in relation to the characterization of the phytochemical profile of SCH, a study that we have started. In attention to your comments, it is mentioned in the discussion (page 11 line 439-441) that the highest percentage of metabolites present in the SCH extract refers to terpenes, flavonoids, saponins and tannins.
- Authors must provide microscopic images of malignancy and normal cell lines following treatments.
We appreciate the suggestion. however, we consider that the cytometry analysis is more conclusive in demonstrating the induction of apoptosis than culture images.
Round 2
Reviewer 3 Report
The authors have made substantial changes in the manuscript, and the quality of the content is significantly improved.
Reviewer 4 Report
The authors have amended the manuscript based on the reviewers comments. I thus recommend prospective publishing in Nutrients Journal.